# Talking to Children and Families about Chronic Pain: The Importance of Pain Education—An Introduction for Pediatricians and Other Health Care Providers

**DOI:** 10.3390/children7100179

**Published:** 2020-10-12

**Authors:** Helen Koechlin, Cosima Locher, Alice Prchal

**Affiliations:** 1Department of Anesthesiology, Critical Care and Pain Medicine, Boston Children’s Hospital, Harvard Medical School, Boston, MA 02115 USA; cosima.locher@plymouth.ac.uk; 2Division of Clinical Psychology and Psychotherapy, Faculty of Psychology, University of Basel, 4055 Basel, Switzerland; 3School of Psychology, University of Plymouth, Plymouth PL4 8AA, UK; 4Department of Psychosomatics and Psychiatry and Children’s Research Center, University Children’s Hospital Zurich, 8032 Zurich, Switzerland; Alice.Prchal@kispi.uzh.ch

**Keywords:** pediatric chronic pain, communication, pain education, pediatricians

## Abstract

Chronic pain in children and adolescents is a common and debilitating health problem. This narrative review will give a brief overview on what pediatric chronic pain is and what treatment options there are for children and adolescents. The specific emphasis will be on pediatric chronic pain education and communication: this narrative review aims to show how important a good patient–health care provider relationship is—it builds the foundation for successful communication—and how this relationship can be established. In addition, we will present five steps that health care providers can perform to explain pediatric chronic pain to patients and their parents and what to keep in mind in their clinical routine. Our review is intended for pediatricians and other health care providers who treat pediatric patients with chronic pain but might feel uncertain on how to best communicate with them.

## 1. Whom Is This Review Intended for?

Pediatricians are usually the first point of contact for children and adolescents with chronic pain and their parents. However, in a recent survey that was sent to all pediatricians in primary care in Switzerland who are registered as working members of the Swiss Society of Pediatrics (SGP), only around 20% of more than 330 respondents had confidence in treating patients with chronic pain [1]. This is alarming, given the high prevalence of chronic pain in children and adolescents, with a significant number reporting moderate to severe restrictions in their daily functioning [2], ranging from lower levels of physical activity [3] to increased absence from school [4] and fewer friends [5,6]. Fast detection, adequate communication, and effective treatment is therefore crucial for patients with chronic pain. Accordingly, this narrative review is meant to introduce pediatricians and other health care providers to chronic pain in children and adolescents, give an overview on current treatment options, and recommend five key talking points to keep in mind when talking to affected families.

## 2. What Is Pediatric Chronic Pain? What Treatment Options Exist?

Up to one in four children will have an episode of chronic pain, i.e., pain lasting for three months or longer or past normal healing time [7], as suggested by systematic estimates [8]. The most common pain syndromes are headache, abdominal pain, and musculoskeletal pain, and girls usually report more pain compared to boys [9]. Pediatric chronic pain is linked to significant psychological, physical, and social concerns for affected children and their families [10,11]. What is more, it also places an enormous burden on healthcare systems—in the United States, chronic pain in adolescents costs around 19.5 billion USD each year and ranks among the most expensive pediatric health problems [12]. Pediatric chronic pain significantly decreases quality of life and is associated with numerous missing days at school or at work for affected children and their parents [13].

In 2015, a Task Force of the International Association for the Study of Pain (IASP) suggested a new and pragmatic classification for chronic pain for the new International Classification of Diseases, 11th Revision (ICD-11). This classification includes seven categories, among them Chronic Primary Pain [7,14]. As the exact etiology of many forms of chronic pain remains unknown, the phenomenological definition of Chronic Primary Pain describes pain in one or more anatomic regions that persists or recurs for more than three months and is associated with significant emotional distress or functional disability [7,15]. This new diagnostic category takes into account that pain is considered an “unpleasant sensory and emotional experience” [16,17], which points to the importance of emotional factors, specifically pain-related distress [18]. The new classification holds the potential for an improvement in pain communication: a JAMA Pediatrics viewpoint [15] suggested the new term Primary Pain for those pain problems for which no “organic lesion” is identified, implying that “pain itself is the disease” and arguing that the patient’s understanding of their pain is closely associated to their compliance with medical advice.

With regard to medical advice, treatment options for pediatric chronic pain include a rehabilitative approach with physiotherapy, occupational therapy, complementary treatments, psychological therapies, and medication. However, current knowledge of treatment approaches is often based on data extrapolated from adults [19,20]. This is particularly problematic for pharmacological treatments, since there is limited evidence that supports their efficacy and safety [20,21] in pediatric patients with chronic pain. In terms of non-pharmacological interventions, psychological therapies usually include multicomponent cognitive behavioral therapy, relaxation-based content, and/or problem-solving strategies. Even though a recent review found some effects of psychological therapies posttreatment, these effects could not be maintained until follow-up [9]. For severely impaired children and adolescents, some hospitals offer intensive interdisciplinary pain treatment, which shows positive preliminary treatment effects in a systematic review of available studies [22]. An overview of the safety and efficacy of all available interventions for chronic pain in children and adolescents relative to each other is currently lacking. Moreover, data suggest that children and adolescents suffering from chronic pain face huge barriers when they search for treatment: a study from the University Children’s Hospital Basel, for example, found that on average, patients reported pain for more than two years prior to their referral to the specialized clinic [23].

When patients finally receive appropriate treatment, they have their own pain concepts that they bring into the treatment situation, and these pain concepts might be more experience-based than evidence-based. Those pain concepts can be associated with common misconceptions. For example, one misconception about chronic pain lays in its differences to acute pain: acute pain functions as a warning signal, such as the pain we feel when placing our hand on the hot stove. Chronic pain, on the other hand, has lost its somatic warning function, it is no longer necessarily a sign of tissue damage or injury [24,25]. However, most patients and their families attribute the signaling function to chronic pain as well, which leads to an increase in worry, fear, and potentially pain-avoidance behavior [26]. A second misconception roots in a traditional and dualistic viewpoint that conceptualizes mind and body as functioning separately and independently [27]. Based on this categorization (that has been challenged since the 1970s [28]), when no underlying tissue damage or injury can be detected, patients and their families are left feeling frustrated and misunderstood and often disengage in new treatment approaches [24]. A third misconception is related to the Chronic Primary Pain classification that subsumes syndromes with no clear organic lesion. Most patients and their families wish for a clear cause for their pain and feel that the experience of pain is only “real” when the pediatrician can provide a clear organic cause for their suffering. What is more, each medical specialty has their own pain diagnosis; neurologists focus on headache, gastroenterologists study abdominal pain, and so on [29]. However, many patients suffer from pain in more than one location or even experience a change of pain location over time [30,31]. Diagnostic uncertainty is therefore a common and burdensome experience for many families. These reasons help to explain why many pediatricians report a lack of confidence when treating patients with chronic pain [1].

We therefore suggest that the very first and probably one of the most important steps of every treatment is a clear and concise explanation of what pediatric chronic pain is—and what it is not. To educate patients and their families about pain can be one component of a comprehensive treatment program or also serve as a stand-alone intervention. Several approaches and modes of delivering pain education have been researched in the past (for a review see [25,32]), proven to be a successful intervention in adults [33], and are often used in children [31], but with a rather limited evidence base in the pediatric pain context so far [34].

Those pain education approaches that have shown to be effective share that they use a biopsychosocial model of chronic pain (see Figure 1). The biopsychosocial model of chronic pain helps to explain how physiologic and psychological factors, and social contexts dynamically interact and contribute to the experience of pain [27,35]. It includes genetic predispositions, central biological, somatic, affective, and cognitive processes that feedback to and receive feedback from the peripheral autonomic, endocrine, and immune systems. These factors all interact, influence, and are influenced by social factors such as interpersonal relationships, family environment, social support or isolation, and previous treatment experiences [36]. The biopsychosocial model is widely accepted as a helpful approach to the understanding and treatment of chronic pain. A clear explanation of this model enables patients and families to better describe what they are experiencing. Importantly, knowing more about the underlying biological, psychological, and social processes of chronic pain might lead to a reconceptualization of pain [37]. Reconceptualization describes the process of the patient understanding that pain is not proportional to tissue damage—especially for chronic pain, as the association between pain and tissue health is weaker the longer pain persists—that pain is influenced by psychological and social factors, and that pain is a subconscious warning of danger of tissue damage, whether the danger is real or not [37,38]. A successful reconceptualization of chronic pain can produce positive effects such as normalizing pain beliefs and attitudes and improving pain and disability outcomes of multimodal interventions [39,40,41]. In a study assessing the feasibility of a brief skills-based group intervention for adolescents with chronic pain and their parents, the neuroscience module (discussion of acute vs. chronic pain, how pain functions in the brain and nervous system, and biologic factors that contribute to the onset and maintenance of chronic pain) led to a significant improvement in knowledge about pain [42].

It is therefore crucial that health care providers help their patients who suffer from chronic pain to learn more about their pain and to deliver those explanations in a fashion that is understandable in order to create positive changes.

## 3. What Can Be Done to Establish a “Good” and Helpful Communication Style Between Patients, Their Families, and Health Care Providers?

Communicating in a helpful manner requires a good relationship between patient, family, and health care provider. When thinking about what can be done to establish a good relationship, research on ‘contextual factors’ in psychotherapy might provide some advice. Psychotherapy research usually aims to examine and uncover the effect of ‘specific factors’, including treatment techniques and specific methods. For multicomponent cognitive behavioral therapy, for example, such specific factors would include exposure, free association, and skills training. Beyond their specific components, however, different psychotherapeutic approaches typically rely on similar ‘contextual factors’ [44], including a positive patient-health care provider relationship, patients’ positive expectations, a plausible treatment narrative, and a powerful treatment ritual (such as going to a clinic, being examined by medical professionals in white coats, and being prescribed medication [45,46,47]). Specifically, research in the field of placebo sheds light on the impact of contextual factors, this is because placebos by definition have no pharmacological ingredients or specific factors, respectively. Relevant placebo effects have been reported in a variety of pediatric conditions including depression and anxiety [48,49], autism [50], attention-deficit/hyperactivity disorder [51], migraine [21,52] and acute pain [53]. Adult placebo studies highlight the importance of a positive patient-health care provider relationship: a sample of patients suffering from irritable bowel syndrome were placed on a waiting list, given placebo acupuncture alone (using a sham acupuncture device), or placebo acupuncture augmented by warmth, confidence, and attention from the health care provider [54]. Across outcomes such as global improvement, symptom severity, and quality of life, the enhanced patient-health care provider relationship provided the most robust effects. A health care provider’s emotionally warm and empathic communication style is therefore crucial to establish a good relationship [55], which in turn makes substantial and consistent contributions to patient outcome [56].

The importance of a plausible treatment narrative, an additional key contextual factor across treatment approaches, has been studied in children: A study with healthy children used a heat-pain paradigm to explore the magnitude of expectation-related placebo analgesia [57]. The instructions in the analgesia-expectation group included a plausible narrative about a child who goes treasure hunting in the desert and uses a lotion in order to protect their hands and feet from being burnt by the hot sand. The applied lotion, however, was a simple hand disinfectant with no anesthetic properties. The control group read an animal book with the experimenter in order to ensure structural equivalence, but with no induction of analgesia expectations. As a result, children in the experimental group exhibited a significantly increased heat pain tolerance and threshold. This result has been replicated in a mixed-age study [58] and points to the potential of a strong treatment narrative in pediatric patients especially: Children are more suggestible compared to adults, and possess great imagination and phantasy [59]. This can be used to create a helpful context for the patient-health care provider communication, such as by using a strong metaphor or empowering the child to develop their own explanations for their pain.

The role of parents and parental behavior in the context of pediatric chronic pain education and communication also needs to be considered. Individual variables (e.g., parental reinforcement, parenting style), dyadic variables (e.g., parent-child interaction, parent-parent interaction), and family level variables (e.g., family environment, overall functioning), possibly mediated or moderated by the child’s age, developmental status, gender, emotional symptoms, coping, and parental pain history, all interact with pain and pain-related disability (which reciprocally influence each other) [60]. This is also reflected in the so-called placebo by proxy effect which occurs when a patient’s (in pediatric populations: a child’s or adolescent’s) response to therapy is affected by the behavior of other people who know that the patient is undergoing therapy (here: the parents) [61].

Parents– despite their best intentions – often engage in unhelpful behaviors such as protective responding to pain or pain catastrophizing [62]. A study found that parents who expressed less confidence in their child’s ability to function despite their pain or who reported more pain-related catastrophizing also showed more protective and monitoring behaviors [63]. Moreover, how much or how little parents should be part of the pain education largely depends on the child’s age and developmental status. In the context of medical decision-making, children around the age of 12 years are considered sufficiently competent for decision-making, while younger children are not thought to be competent enough to act for themselves [64,65]. Four components are critical: a child needs to be able to express a choice, to understand the information provided, to reason, and to appreciate the relevance of the situation [66]. These components can serve as a valid starting point when evaluating if and how much parents should be involved in the pain education sessions and in treatment in general. However, health care providers should also keep potential worries and fears of parents in mind: Parents need to understand and support the pain concept that is developed in pain education sessions in order not to involuntary disrupt the therapeutic process [60].

To sum up, what can we learn from research on contextual factors and pain education that might help health care providers communicate better with their pediatric patients suffering from chronic pain? First, in order to obtain a good patient-health care provider relationship, the patient’s feeling of being understood and cared for, and an emotionally warm and empathic communication style of the health care provider are essential. Second, the patient’s treatment expectations need to be taken into account: Positive, albeit realistic expectations that are grounded in a sound and meaningful treatment rationale can help to increase the likelihood of treatment success. Third, the patient needs to be provided with understandable and plausible information that allow for a reconceptualization of chronic pain. The use of an evidence-based model such as the biopsychosocial model of chronic pain and a delivery mode that includes metaphors, drawings, and layman terms is highly recommended. Fourth, as the young patient is embedded in a family system and parental behaviors and beliefs critically influence the patient’s outcomes, the parents need to be involved in the process of understanding their child’s pain better. This also allows them to adjust their attitudes and potentially correct unhelpful behaviors towards the child. In addition, depending on the developmental status of the child, parent involvement can be regulated.

## 4. How Can Pediatric Chronic Pain Be Explained? How Should Pain Education Be Performed?

Pain education is recommended as an important part of the interprofessional treatment approach for pediatric chronic pain, aiming to improve pain and function by teaching the relevant biopsychosocial mechanisms of pain [67,68]. Educational approaches aim to provide a framework to understand the individual pain condition, by illustrating what pain is, what function it serves, and how it works [25].

There are several expressions for pain education as a specific intervention, including “pain neuroscience education”, “therapeutic neuroscience education”, and “Explaining Pain” [32]. Unfortunately, and despite improvements in the last years, resources for pain education for health care professionals across all disciplines are still limited [69].

Children and their parents strive for an explanation for their pain. Understanding more about the mechanism of pain can help to reduce the threatening nature of pain and enable patients to change their behavior if necessary. While we usually have suitable biological explanations for acute pain, the provision of a thorough understanding in chronic pain is more complicated. Over the last decades, scientific knowledge about chronic pain has grown, allowing to understand this complex and biopsychosocial phenomenon better – but how shall we explain this to a teenager or a young child in pain and their concerned parents? The following five steps are a possible starting point for pain education in clinical practice (see Figure 2).

Initially, health care providers should ask about the current understanding and concept of pain of the family. The idiosyncratic explanation of pain – of child and parents alike – should be the basis on which to build pain education on, since an individual’s interpretation of a pain problem can influence the pain itself and pain-related behavior, and a thorough understanding of a child’s previous experiences and knowledge is important to facilitate pain education [70]. Pain education is only successful when a common denominator between the subjective concept of the child and the health care provider’s scientific knowledge about the biopsychosocial nature of chronic pain can be found [71]. Therefore, the crucial first step is to thoroughly explore the pain concept of the patient and their family.

In a second step, health care providers should discuss with the family what is known or not known about possible biomedical causes of the pain in the particular case. Since almost all patients with chronic pain have a history of extensive medical search for physical causes, it is important to address this topic, summarize findings, and especially ask parents about further doubts and worries. At this point, health care providers very often need to clarify that the pain might also be unrelated to tissue damage and this sort of pain is just as “real” as any other type of pain. 

In a third step, the scientific knowledge about pain can be deepened; this is where the actual pain education takes place. Using a modified Delphi process, an interdisciplinary group of health care providers and researchers recently identified and gained consensus on seven key learning objectives for adolescent pain science education: (1) Pain is a protector; (2) The pain system can become overprotective; (3) Pain is a brain output; (4) Pain is not an accurate marker of tissue state; (5) There are many potential contributors to anyone’s pain; (6) We are all bioplastic and; (7) Pain education is treatment [72]. Based on these learning objectives, patients should be able to understand that pain serves as a protector and is therefore our lifesaving warning system in many cases [73]: Acute pain protects the organism from harm by alerting us whenever there seems to be threat to our health and well-being. But when pain persists, the pain system can become more and more sensitive and tends to be overprotective by nature [74]. Chronic pain has lost its protective function then, nerves can become hypersensitive to stimuli, and pain is present even if there is no evident threat to the body [75]. This is why rest is usually not helpful for patients with chronic pain – on the contrary, an active therapy approach is necessary. Further, patients need to know that the experience of pain is always a product of our brain. Pain is not created in the tissues, but rather is a conscious feeling that urges us to act to protect a particular part of the body [74]: The brain has to interpret the sensory input arising from the nerves through the spinal cord and in this process, prior experiences, thoughts and emotions are significantly involved. Therefore, pain is not an accurate marker of tissue state. In the whole pain network, there are many potential contributors to anyone’s pain that offer numerous ways how pain can be controlled or modulated. Pediatric patients with chronic pain and their parents need to understand that pain is a biopsychosocial phenomenon and factors like emotional state, previous exposure, attention, learning, understanding of pain, context, sleep, nutrition, physical state, and other sensory cues may significantly influence the individual pain experience [76]. At this point it is important to clarify that the etiology of chronic pain cannot be reduced to either physical or psychological – it is always a far more complex [15].

In a fourth step and based on the pain-related knowledge previously taught, patients and their parents should be given a treatment rationale. They should understand that we are all bioplastic and pain can be reduced by an active therapy approach. Given the biopsychosocial nature of chronic pain, patients and families can be introduced to a multidisciplinary treatment approach to pediatric chronic pain that often includes physical, psychological, social, and sometimes pharmacological interventions.

Finally, the fifth step refers to the transfer of knowledge to the social environment of the patient. Once patients and parents have gained in-depth understanding of the “what, why and how of pain” [70], health care providers should discuss with the family if there is any need to pass on relevant information about the child’s pain and treatment to other important people in the patient’s social environment. Chronic pain in young people can negatively affect relevant life domains such as school attendance and peer relationships [4,77], and some adolescents are under suspicion that they are “faking” their pain symptoms, which are by their nature not visible to others [78]. Therefore, addressing misconceptions about chronic pain held by teachers, coaches, peers, or other relevant persons might help to improve patient’s well-being.

## 5. Which Methods Are Helpful for Delivering Pain Education?

There are several methods of how pain education can be delivered of which verbal instruction is the most common. Oral presentations can be accompanied by quick sketches or slides to help further solidify verbal expiations. Often used visualizations are the three circles of the biopsychosocial model of pain (Figure 1) or a picture of how pain is being transmitted through the spinal cord and into the brain.

Analogies and metaphors can also help to understand the pain phenomenon. Performing pain education by using metaphors has shown to be helpful in increasing pain knowledge in an adult population [41]. In the case of fibromyalgia, for example, the body has been described as a “very clever computer” where pain is caused by a software rather than a hardware problem. The software problem is due to the adaption of the body, i.e., when people have to “keep going” despite “stop signals” such as pain and fatigue [79]. For the pediatric field, Coakley and Schechter [24] provided a valuable collection of metaphors and analogies for health care providers to explain onset, maintenance, and treatment of pediatric chronic pain. For example, to explain pain sensitivity, they compare persistent pain with a car alarm which sometimes goes off even though there is no sight of danger. Pain rehabilitation is described as an “athlete’s recovery program” rather than a sick person waiting for a cure. To explain mechanisms of pain transmission and pain control, the “pain gate”, originally based on the gate control theory [80], can also serve as a helpful analogy (Figure 3). This virtual “gate" gives an example of how a pain signal goes through various “gates” which can be closed (and therefore reduce the pain) by sensory inputs as well as cognitive-behavioral methods [81].

In addition to one-to-one conversation, online educational videos or online therapy sessions provide an ideal opportunity to engage youth in pain education [83]. Young people are growing up in an advanced technological age and are using digital media to answer their health questions [84]. As the quality of online information is not always sufficiently good, it is advisable to recommend helpful and scientifically sound sources for educational videos and discuss the content with patients. In a systematic review about pain education videos on YouTube, Heathcote and colleagues [85] identified over one hundred pain education videos in English, some of them of good quality and useful for delivering pain education in clinical practice.

## 6. What Is Important to Keep in Mind in the Clinical Setting?

Pediatric pain education can be performed by any professional of the health care team who has the required knowledge. In order to establish good knowledge about pediatric chronic pain, physicians, nurses, psychologists, physiotherapists, and other allied health professions have to work together and recognize each other’s specific part in pain education and management. In addition, it is important to keep in mind that many patients who present with chronic pain also report symptoms of anxiety or depression, and have a history of psychological trauma or adverse childhood experiences [86,87,88,89]. Symptoms that are associated with posttraumatic stress disorder have a high prevalence in youth with chronic pain – and in their parents [90]. Hence, having an interdisciplinary team of professionals who share a common understanding of pediatric chronic pain and is familiar with psychiatric disorders and diagnostics is crucial to provide the best care possible for this patient group that often presents with a range of symptoms and comorbidities.

Oftentimes, communication about pain does not happen in a structured program or in a standardized setting, but is part of normal clinical routine within a private practice or a hospital. Ideally, pain education should be available everywhere, but there are some aspects to consider in everyday clinical practice:

When discussing pain with children, it is especially important to adapt communication to the cognitive capacities [32] and to carefully consider language [91]. In addition, the duration of conversation should be considered, and shorter sessions are recommended for younger children.

Then, parents and children have to be equally involved in the pain education process, since parents’ beliefs about the etiology of the child’s pain and their behavior crucially influences the child’s pain experience [60]. 

Further, clinical practice shows that education is a continuous process and oftentimes needs repetition. By discussing how chronic pain can exist despite no traceable tissue damage, the pain expert must possibly try to undo patients’ and families’ intuitive understanding of pain, which stems from numerous experiences with acute pain and renders chronic pain difficult to understand [24]. Pain education is therefore usually not a one-time conversation, but needs repetition and a certain power of persuasion. However, and noteworthy, research shows that even a single education session on the neurophysiology of pain can lead to a decrease in subsequent pain ratings and an increase in pain knowledge [33].

It is also important to keep in mind that some of the treatment recommendations for pediatric patients with chronic pain may sound counterintuitive for children and parents [24] and do not correspond to the common conception of “treating the direct cause of the problem”. For example, suggesting increased activity for patients with chronic pain does not imply that insufficient activity was responsible for the pain problem. Similarly, the offer of psychological support to learn strategies to better cope with pain does not mean that the pain problem is solely due to psychological reasons. This makes it all the more essential to provide clear and in-depth information to children and family and to allow for time to respond to questions and concerns.

Importantly, we have to keep in mind that most patients who go through diagnostic procedures see many different health care professionals before they are being referred to a specialized institution. However, gathering information and building or (re-)conceptualizing pain concepts already starts with the first contact with a health care provider. Therefore, and ideally, all professionals in an institution should have at least some basic knowledge about pediatric chronic pain and a consensus about the wording they use in communicating about pain. It should be an important goal to make all patients feel valued and taken seriously, and for them to receive a plausible explanation for their individual pain experience – irrespective of the existence of an organic lesion. Based on our knowledge of the complexity and nature of chronic pain, all health care providers must represent this attitude and comments like “you don’t have anything” or “it is all psychological” should absolutely be avoided. The ideal combination to facilitate pain education and communication is therefore a solid scientific understanding of chronic pain paired with an empathic and warm communication style that respects and values the individual patient and their family.

## Figures and Tables

**Figure 1 children-07-00179-f001:**
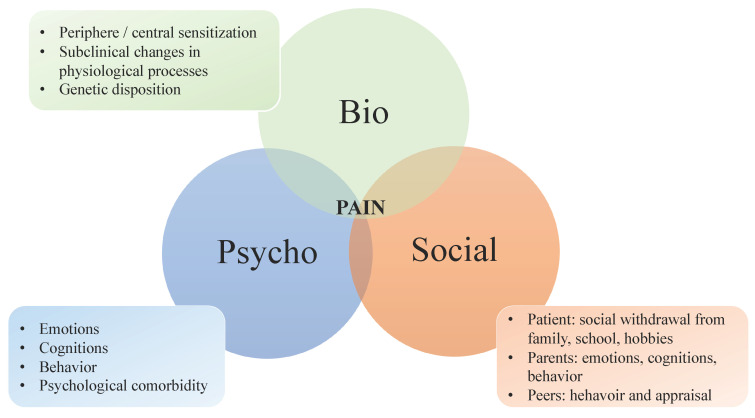
Biopsychosocial model of pain. Based on Wager & Zernikow, 2014 [43].

**Figure 2 children-07-00179-f002:**
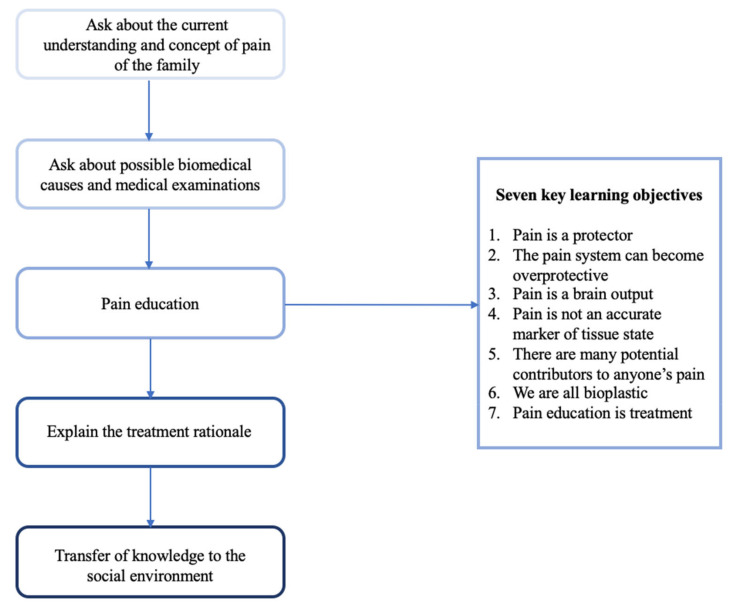
Flow chart: five suggested discussion points in clinical practice.

**Figure 3 children-07-00179-f003:**
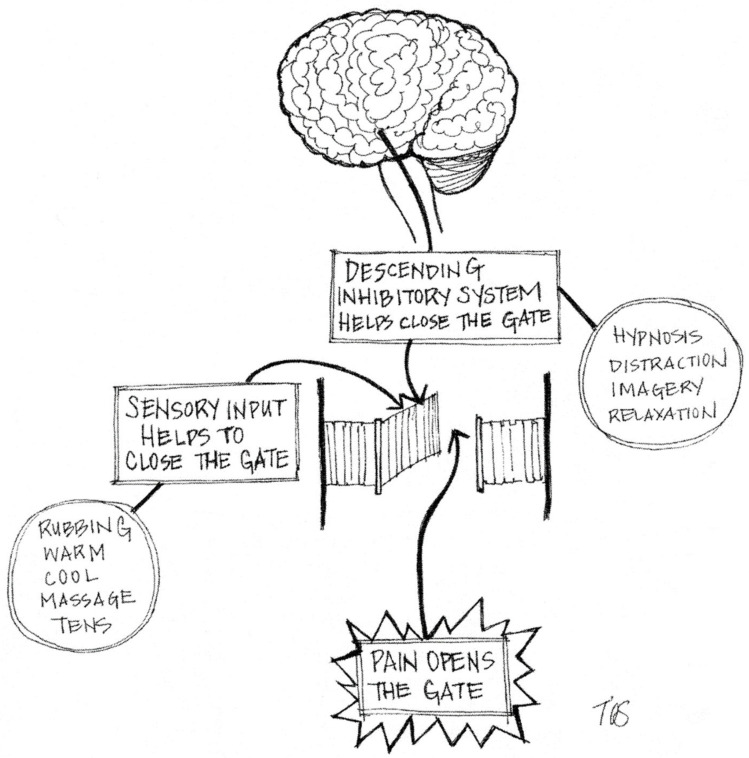
The pain gate [81], based on Wall and Melzack, 2008 [82]. © Leora Kuttner [81], used with permission.

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
