# Peer review of "Talking to Children and Families about Chronic Pain: The Importance of Pain Education—An Introduction for Pediatricians and Other Health Care Providers"

_children, 2020, doi:10.3390/children7100179_

Round 1

Reviewer 1 Report

good concept identified, and important part of understanding persistent (chronic) pain;

2 points:

1, In research findings have suggested the use of the word persistent rather than chronic 

2, research has also found that this concept of pain language and explanation has shown to decrease the effect and intensity of "perceived" pain, studies by D Butler has found the understanding of pain has a direct relationship on how the individuals experiences pain

Author Response

In research, findings have suggested the use of the word persistent rather than chronic

Answer: We thank the reviewer for pointing to this important topic. We agree that the decision of which terms to use in the context of health care should not be taken lightly and that for some people, the term “chronic pain” refers to a lifelong illness that cannot be cured. Other patients might constantly try to fix “chronic pain” because they conceptualize it as something they need to cure and then be done with. However, in order to be in accordance with the new ICD-11 diagnostic entity, we decided to use the term “chronic pain” rather than “persistent pain”.

Research has also found that this concept of pain languages and explanation has shown to decrease the effect and intensity of “perceived” pain, studies by D. Butler have found the understanding of pain has a direct relationship on how the individual experiences pain.

Answer: We thank the reviewer for this comment. We have now added one sentence on p. 8, lines 343-345, citing a reference by Louw, Diener, Butler and Puentedura (2011), stating: However, and noteworthy, research shows that even a single education session on the neurophysiology of pain can lead to a decrease in subsequent pain ratings and an increase in pain knowledge [86].

Reviewer 2 Report

The current study aimed to provide an overview of pain education for pediatricians and primary care providers. Overall, it is extremely well written and will be a nice resource for non-pain providers. Several important considerations for pain education were also addressed. However, authors may consider adding an additional point of discussion to optimize recommendations and care for these families. Most notably, strong emerging research suggests that a significant subset of families with chronic pain (either parent or child or both) hold a history of psychological trauma or adverse childhood experiences (ACEs). In this context and given that (as authors point out) these providers are often the "first line of defense" in recommending care for families with pediatric pain, authors are encouraged to consider adding discussion in this manuscript on the importance of trauma-informed care as a point of education for these providers to optimize treatment or at the very least, an avenue for future investigation. 

Author Response

The current study aimed to provide an overview of pain education for pediatricians and primary care providers. Overall, it is extremely well written and will be a nice resource for non-pain providers. Several important considerations for pain education were also addressed. However, authors may consider adding an additional point of discussion to optimize recommendations and care for these families. Most notably, strong emerging research suggests that a significant subset of families with chronic pain (either parent or child or both) hold a history of psychological trauma or adverse childhood experiences (ACEs). In this context and given that (as authors point out) these providers are often the "first line of defense" in recommending care for families with pediatric pain, authors are encouraged to consider adding discussion in this manuscript on the importance of trauma-informed care as a point of education for these providers to optimize treatment or at the very least, an avenue for future investigation. 

Answer: We thank the reviewer for this positive feedback and agree that adverse childhood experiences or trauma oftentimes play an important role when working with families affected by pediatric chronic pain. We have now added the following sentences on p. 11: In addition, it is important to keep in mind that many patients who present with chronic pain also report symptoms of anxiety or depression, and have a history of psychological trauma or adverse childhood experiences [85–88]. Symptoms that are associated with posttraumatic stress disorder have a high prevalence in youth with chronic pain – and in their parents [89]. Hence, having an interdisciplinary team of professionals who share a common understanding of pediatric chronic pain and is familiar with psychiatric disorders and diagnostics is crucial to provide the best care possible for this patient group that often presents with a range of symptoms and comorbidities.